# MedCaseReasoning: Evaluating and learning diagnostic reasoning from clinical case reports

## Abstract

Doctors and patients alike increasingly use Large Language Models (LLMs) to diagnose clinical cases. However, unlike domains such as math or coding, where correctness can be objectively defined by the final answer, medical diagnosis requires both the outcome and the reasoning process to be accurate. Currently, widely used medical benchmarks like MedQA and MMLU assess only accuracy in the final answer, overlooking the quality and faithfulness of the clinical reasoning process. To address this limitation, we introduce `MedCaseReasoning`, the first open-access dataset for evaluating LLMs on their ability to align with clinician-authored diagnostic reasoning. The dataset includes 14,489 diagnostic question-and-answer cases, each paired with detailed reasoning statements derived from open-access medical case reports. We evaluate state-of-the-art reasoning LLMs on `MedCaseReasoning` and find significant shortcomings in their diagnoses and reasoning: for instance, the top-performing open-source model, DeepSeek-R1, achieves only 48% 10-shot diagnostic accuracy and mentions only 64% of the clinician reasoning statements (recall). However, we demonstrate that fine-tuning LLMs on the reasoning traces derived from `MedCaseReasoning` significantly improves diagnostic accuracy and clinical reasoning recall by an average relative gain of 29% and 41%, respectively.

## 1 Introduction

Large language models (LLMs) are increasingly deployed in medicine (AMA, 2025), where they show promise in performing complex tasks like clinical reasoning (Goh et al., 2024) and disease diagnosis (McDuff et al., 2025). While there are a number of benchmark datasets designed to assess the diagnostic capabilities of medical LLMs (e.g. MedQA (Jin et al., 2020), MMLU (Hendrycks et al., 2020), MultiMedQA (Singhal et al., 2023), and MedXpertQA (Zuo et al., 2025)), these datasets all share the same limitation in that they only assess the correctness of the model's final answer. Unlike domains like mathematics (Ahn et al., 2024) or coding (Ding et al., 2024), where the reasoning process is secondary to the correct answer, medical diagnoses require both the final diagnosis and the thought process to be defensible. In clinical practice, physicians need to articulate coherent rationales, which is important for ensuring informed consent, meeting the standard of care, facilitating documentation and billing, and enabling clinical collaboration (Patel et al., 2005). Mistakes or deficiencies in reasoning, even if the correct diagnosis is given, can risk case mismanagement. Highlighting this gap, a recent study found that even frontier models like GPT-4 could produce the correct diagnosis for incorrect reasons in up to a third of clinical scenarios (Jin et al., 2024). Thus, evaluating and improving model capabilities to reason and answer correctly is crucial for LLMs to gain clinical viability and trustworthiness.

In this work, we propose `MedCaseReasoning`, a diagnostic reasoning dataset with two complementary goals: (1) an evaluation benchmark to assess how well LLM reasoning aligns with clinician-authored diagnostic reasoning, and (2) a training dataset for improving diagnostic reasoning in LLMs from reasoning traces. This clinician-validated dataset is created from publicly available case reports published in PubMedCentral (PubMed Central). We perform filtering and clinician validation from an initial set of 98,994 case reports to produce a corpus of 14,489 diagnostic QA, with a high-quality test subset of 897 cases. First, we

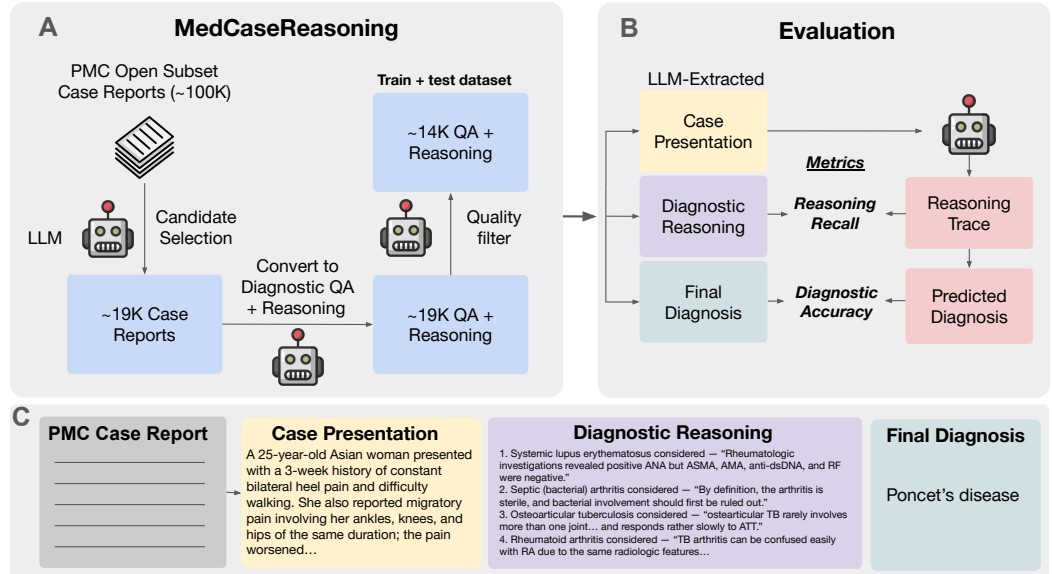

Figure 1: A schematic of the `MedCaseReasoning` case processing pipeline. **A**: From the initial set of 98,994 PMC Open Subset Case Reports, we select 28,313 appropriate candidate cases, which are converted to QA format and then filtered for quality. The resulting 14,489 cases form the `MedCaseReasoning` dataset, with 897 test cases and 13,092 training cases. A full example is available in Table 8. **B**: For each test case, the case presentation is posed to an LLM to reason and then answer with a diagnosis. The clinician-authored diagnostic reasoning is compared with the LLM-generated reasoning to produce the reasoning recall score, and the case diagnosis is compared with the predicted diagnosis to produce diagnostic accuracy. **C**: The original case report text is converted into three sections: the case presentation that contains the relevant and sufficient patient information for making a diagnosis; the diagnostic reasoning that includes the diagnostic decision-making by the case author (in enumerated statements); and the final diagnosis.

observe that `MedCaseReasoning` is a meaningfully difficult and unsaturated benchmark, with the top commercial model, *OpenAI o3*, achieving only 65% on 10-shot diagnostic accuracy. Second, we find that even top-performing reasoning models are substantially limited in their ability to reason like clinicians, where the best open-source model, *DeepSeek-R1*, only recalls around half of the reasons provided in the case report. Third, we find that training on `MedCaseReasoning` can significantly improve the diagnostic and reasoning capabilities of LLMs: both diagnostic accuracy and clinician reasoning alignment increased after fine-tuning three top medical open-source LLMs, with an average relative gain of 29% and 41%. respectively. Importantly, after training on `MedCaseReasoning`, their performance on NEJM CPC, a held-out test dataset, also improved, showcasing the value of `MedCaseReasoning` in improving diagnostic generalizability.

**Main Contributions**:

- We release `MedCaseReasoning`, an open-access dataset of 14K+ clinical diagnostic cases from 800+ medical journals and 30+ specialties with accompanying clinician-authored reasoning.

- To produce `MedCaseReasoning`, we describe a scalable, multi-step, clinician-validated pipeline for converting raw case reports into high-quality QA-format diagnostic cases.

- We evaluate state-of-the-art reasoning LLMs models on `MedCaseReasoning`, which reveals limitations in medical diagnostic reasoning.

- We show training on dense reasoning traces from `MedCaseReasoning` improves the diagnostic accuracy and reasoning recall of open-source LLMs.

Our work is the first medical benchmark dataset that explicitly evaluates the reasoning accompanying a diagnosis by comparing it to real-world case reports written by clinicians. These cases contain relevant differential diagnoses along with case presentations that present sufficient detail for an accurate diagnosis. Our findings underscore the significant challenges of aligning the reasoning of current top-performing LLMs with established clinical practice and provide a dataset for improving this alignment.

**Related Works** While commonly used medical benchmark datasets like MedQA (Jin et al., 2020), MMLU (Hendrycks et al., 2020), and MedXpertQA (Zuo et al., 2025) include diagnostic reasoning questions, the clinical cases presented are typically shorter, textbook-style cases intended to evaluate students and test medical knowledge, not real-world clinical cases that physicians may encounter. Several datasets construct clinical vignettes from real-world sources, such as PubMedQA (Jin et al., 2019), which generated QA examples based on PubMed articles, and MedAlign (Fleming et al., 2023), which generated clinical vignettes from electronic health records. By comparison, MedCaseReasoning is derived directly from clinician-authored case reports, which are valuable tools for advancing clinical practice (Nayak, 2010).

The New England Journal of Medicine Clinicopathologic Conferences (NEJM CPC) is a collection of case reports sourced from Massachusetts General Hospital in Boston, MA (McDuff et al., 2025). MedCaseReasoning improves upon this dataset in several ways. First, the collection of case reports from PubMedCentral represents a globally diverse set of clinician and patient backgrounds, whereas NEJM CPC represents the medical practices of physicians from a single hospital system on a specific population. Second, the NEJM CPC dataset consists of only 302 test cases, while MedCaseReasoning is comprised of over 14K cases, allowing for more in-depth analysis as well as model fine-tuning. Third, MedCaseReasoning is derived from open-access articles on PubMedCentral, whereas NEJM CPC is available only under license.

Several studies (Goh et al. (2025) and Strong et al. (2023)) have employed grading rubrics applied to curated clinical vignettes, but are restricted to a handful of cases, not openly accessible, and focus on human-AI comparison rather than establishing benchmarks for standalone LLM reasoning performance. Studies by Kanjee et al. (2023), McDuff et al. (2025), and Gemini (2023) leveraged cases from NEJM CPC to evaluate model capabilities, but primarily focusing on generating differential diagnosis (DDx) lists, not full reasoning traces.

Recent models like HuaTuoGPT-o1 (Chen et al., 2024) and MedReason (Wu et al., 2025) involve supervised fine-tuning (SFT) on diagnostic reasoning traces, but the traces are distilled from larger proprietary models (e.g., GPT-4) rather than being grounded in reasoning authored by clinicians based on real patient cases. One study by (Savage et al., 2024) conducted manual clinician evaluations of reasoning traces from GPT-3.5 and GPT-4 to identify logical inconsistencies, but such methods are labor-intensive and inherently limited in scale, making them infeasible for large-scale benchmarking. Collectively, prior work highlights a gap in large-scale, open-access benchmarks grounded in real-world, clinician-authored reasoning that evaluate the complete diagnostic thought process beyond just final answer accuracy.

## 2 METHODS

### 2.1 DATA CURATION

Case reports are typically written to highlight a rare or complex disease. A core tradeoff exists between the *novelty* and *usefulness* of a case report – models should learn from difficult cases, but should be able to practically deduce the diagnosis from the presented information. To this end, we produced a LLM-based pipeline with the goal of maximizing case novelty and usefulness. As expert verification on each step is prohibitively expensive, we perform our clinician validation only on the outputs of the last step.

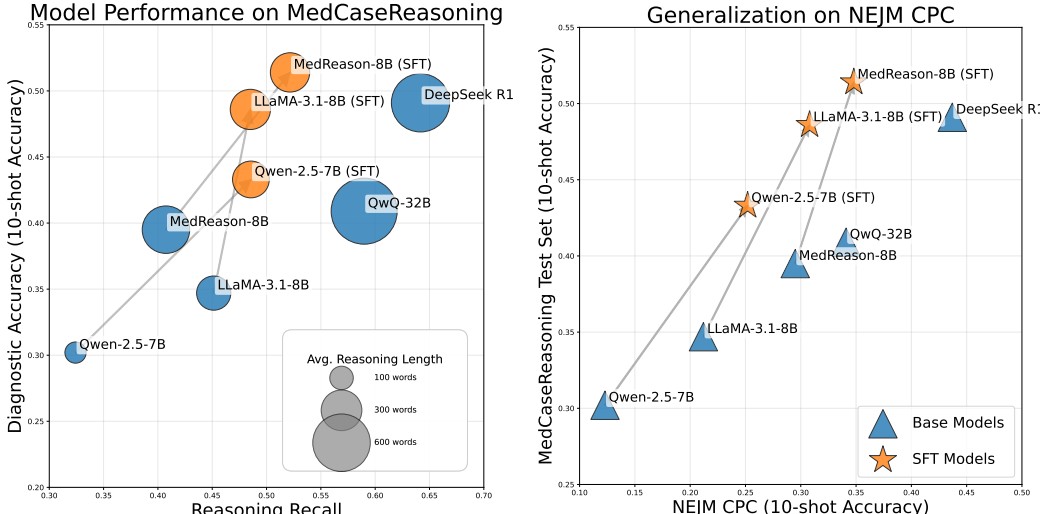

Figure 2: (Left) Evaluation of LLMs on the `MedCaseReasoning` test dataset (N = 897), along with diagnostic accuracy and reasoning recall percentage. (Right) Model performance on the `MedCaseReasoning` test set and NEJM case studies (N = 302), showing a strong correlation between the two benchmarks. In both plots, accuracy is computed with 10-shot accuracy; circle size encodes the average length of the model-generated reasoning trace. LLaMA-3.1-8B and Qwen-2.5-7B are both Instruct variants (omitted for brevity).

| Dataset | $N$ | Open Access | Reasoning | Question Type | Question Source |
|---|---|:---:|:---:|---|---|
| MedQA (Jin et al., 2020) | 12,723 | ✅ | ❌ | Clinical vignettes | USMLE |
| MedXpertQA (Zuo et al., 2025) | 1,050 | ✅ | ❌ | Clinical vignettes | Licensing exams |
| NEJM CPC (McDuff et al., 2025) | 302 | ❌ | ❌ | Real diagnostic cases | NEJM CPC |
| **MedCaseReasoning** | 14,489 | ✅ | ✅ | Real diagnostic cases | PMC Case Reports |

Table 1: Comparison of clinical diagnosis benchmark datasets. `MedCaseReasoning` is the only and largest open-access dataset with clinician reasoning. Existing datasets do not contain expert reasoning.

1. **Data Provenance:** We started with the PMC Open Subset pulled from Jan 1st, 2005-April 27th, 2025. The query was filtered for free-access articles with article type "case report", which yielded 98,994 case reports. The article's text was extracted from the main body in the XML file, along with the PMCID, publication date, title, journal name, and article link.

2. **Candidate Selection:** Next, we excluded case reports without the word *differential* (ie. a discussion on differential diagnoses), which reduced the set to 28,313 case reports. These initial candidate case reports were converted into diagnostic QA with *o4-mini*, a strong reasoning model that can be run at scale. The full prompt is available in the Appendix (Prompt 1) and instructs models to identify the core conflict within the case report and provide an information cutoff that would ensure the presented information does not give away the final diagnosis. In parallel, each candidate case report was scored according to the following criteria: 1) thoroughness of case presentation, 2) presence of explicit differential diagnosis, 3) dependence on integrative clinical reasoning, 4) transparency of diagnostic reasoning process, and 5)

| Reasoning | LLaMA-3.1-8B-Instruct (Base) | LLaMA-3.1-8B-Instruct (SFT on MedCaseReasoning) | Groundtruth (from MedCaseReasoning) |
|---|---|---|---|
| **Trace** | The patient has a history of Wilson disease , a genetic disorder that leads to copper accumulation in the body. However, the brain MRI shows no evidence of pathological copper deposition , and laboratory tests show normal copper metabolism – suggesting the current condition is not related to Wilson disease . The symptoms … are consistent with a psychotic disorder. The history of Wilson disease and orthotopic liver transplantation may have led to post-liver-transplant psychosis … Given the patient's symptoms and history, post-liver-transplant psychosis is the most likely diagnosis. | We first considered acute graft rejection due to non-adherence with tacrolimus and mycophenolate, but this was ruled out because tacrolimus trough levels were low. Next, tacrolimus neurotoxicity was excluded for the same reason. We also considered copper overload from Wilson disease , but MRI showed no pathological copper deposition . Delirium was excluded; finally, we evaluated substance-induced psychosis but found "no other medical condition or substance use disorder ." With alternatives excluded, schizophrenia was suspected. | 1. Wilson-disease recurrence — "no abnormalities, including pathological copper accumulation … no evidence of abnormal copper metabolism … psychosis due to recurrence of Wilson disease was ruled out." 2. Tacrolimus-induced psychosis — " Tacrolimus-induced psychosis was ruled out because the trough tacrolimus concentration had not increased before onset of psychosis, and psychosis persisted after the trough value decreased." 3. Substance-related or other medical causes — "No evidence of other medical conditions or a substance use disorder that could cause psychotic symptoms was found." |
| **Answer** | Post-liver-transplant psychosis | Schizophrenia | Schizophrenia |
| **Coverage** | **1/3 reasons recalled** | **3/3 reasons recalled** | **Groundtruth** |

Table 2: An example of comparing reasoning traces against the case report reasoning. The reasoning from the case report is extracted using an LLM as an enumerated list. Then, each element of the list is referenced against the model's reasoning trace, which gives us the "reasoning recall" for a given trace.

presence of stated final diagnosis. These criteria were developed by clinicians, and *o4-mini* was used to produce the actual scores. Criteria 1, 3, and 4 are given out of 1-5 points, whereas criteria 2 and 5 are yes/no responses. We filtered out cases where 1) thoroughness of case presentation was either 1 or 2 (seriously deficient or major gaps), and also removed cases that either did not discuss 2+ plausible alternatives or did not state a final diagnosis. The full prompt is available in the Appendix (Prompt 2). After filtering, there remained 19,428 total case reports.

3. **Quality Filter:** Next, to avoid model blind spots, we checked each of the generated case reports using a separate LLM (*gemini-2.5-pro*), which evaluated the generated case report's faithfulness to the source article and the plausibility of each case report. The prompt is also available in the Appendix (Prompt 3). We removed cases that had any flags, leaving a final total of 14,489 cases. We created an initial testing subset of 897 cases where the transparency and integrative diagnostic reasoning scores were at least 4 or 5.

The case reports used in MedCaseReasoning span over 800 different medical journals, with the top 10 shown in Table 5. Additionally, the diagnostic case prompts from MedCaseReasoning are significantly longer and more detailed than the ones from MedQA (displayed in Figure 3). Furthermore, case report publication dates are heavily enriched towards years after 2020, with over 16% after Jan 1 2024 (Figure 4). Additionally, our pipeline can be regularly updated, incorporating new case reports with low marginal cost.

To validate the case prompts, diagnostic reasoning, and final diagnosis extracted from each case report, a team of four board-certified physicians reviewed a total of 100 randomly selected cases. For each case, they responded to three questions related to the presence of hallucinations, faithfulness, and reasonableness of each case. The full questions can be found in the Appendix (Table 5).

## 2.2 EVALUATING DIAGNOSTIC ACCURACY IN MODELS

We evaluate each model's diagnostic correctness using LLM-as-a-judge, in accordance to previous literature (e.g., McDuff et al. (2025); Wu et al. (2025)). We adopt the same prompt used in McDuff et al. (2025), (Prompt 7), which has been validated to have high concordance with human raters, and use *gpt-4o-mini* as the judge model for its speed and accuracy.

Differential diagnoses typically contain between five to ten candidate diagnoses which are followed-up in a clinical setting. Also following evaluation schema from McDuff et al. (2025), each model is evaluated a total of 10 times (temperature of 0.8 and top-p of 0.95), and the N-shot performance is recorded.

As an external validation to the `MedCaseReasoning` Test Set, we also evaluate models on case reports from NEJM Clinicopathologic Conferences. In particular, we use a subset of 302 cases from previous works (McDuff et al., 2025; Kanjee et al., 2023; Gemini, 2023) as a gold-standard dataset of complex diagnostic cases. We manually extracted each case report with a personal license and fed the case presentation portion as the case prompt without any additional modifications. As NEJM CPC is not an open-access publication, we are not able to open-source this validation set. Additionally, no NEJM case reports were included in `MedCaseReasoning` or used in training any models.

### 2.3 Evaluating Reasoning Traces in Models

While some case studies include comprehensive reasoning steps, it is not guaranteed that case reports include an exhaustive set of reasoning steps. As such, we focus our study on evaluating how well models *recall* clinician-produced reasons (as opposed to evaluating *precision*). We define this metric as the "Reasoning Recall". (Note: We were not able to evaluate reasoning recall for *OpenAI o3* and other frontier models where the full reasoning traces are not available via API call. While they can produce reasoning tokens when prompted (i.e. with <think> tags), the comprehensive reasoning that is used to inform a diagnosis is not accessible.)

**Definition 1 (Reasoning Recall)** *Let $N$ be the total number of cases. For each case $i$, let*

$$R_i = \{groundtruth\ reasons\ from\ case\ report\}, \quad T_i = \{reasons\ from\ model\ reasoning\ trace\}.$$

*For case $i$, the recall rate is*

$$c_i = \frac{|R_i \cap T_i|}{|R_i|}.$$

*The* Reasoning Recall *is the average of these rates:*

$$\mathrm{RR} = \frac{1}{N} \sum_{i=1}^{N} c_i = \frac{1}{N} \sum_{i=1}^{N} \frac{|R_i \cap T_i|}{|R_i|}.$$

For example, a reasoning recall of 50% indicates that on average, a model's reasoning trace will cover half of the reasons provided by the case report. We evaluate the reasoning trace associated with its best-of-10 response, where we use the trace from a randomly selected correct response (or incorrect if there are no correct responses). The evaluation is implemented using an LLM-as-a-judge, where *o4-mini* is instructed to return a JSON with a decision on whether each of the groundtruth reasons are found in the reasoning trace. We validate this step with annotations from a board-certified physician. The prompt used is found in the Appendix (Prompt 5).

### 2.4 Supervised Fine-Tuning on Case Report Reasoning

In addition to evaluating model reasoning traces, we explore whether fine-tuning on extracted reasons directly can improve model performance on diagnostic accuracy and reasoning recall. One technical challenge is that the extracted diagnostic reasoning is formatted as an enumerated list of summary points and quotations, rather than a cohesive reasoning trace. Naturally, we can use LLMs to "stitch" the list of points into reasoning traces without adding new information. We provide the prompt used to perform this step in the Appendix (Prompt 4). The stitching process may introduce biases if a stronger model introduces its own priors into the reasoning traces. To control for this, we ensure that the model that is being fine-tuned is also the model that stitches the reasoning traces together. An example of a stitched reasoning trace can be found in the Appendix (Prompt 5).

| MedCaseReasoning Test Set | | | | |
|---|---|---|---|---|
| *Base Models* | *Reasoning Recall* | *1-shot Acc.* | *5-shot Acc.* | *10-shot Acc.* |
| OpenAI o3 | N/A | $\mathbf{0.470}_{0.440-0.500}$ | $\mathbf{0.609}_{0.579-0.639}$ | $\mathbf{0.645}_{0.618-0.675}$ |
| DeepSeek R1 | $\mathbf{0.642}_{0.616-0.667}$ | $0.320_{0.291-0.349}$ | $0.447_{0.417-0.478}$ | $0.480_{0.450-0.510}$ |
| QwQ-32B | $0.590_{0.560-0.619}$ | $0.272_{0.245-0.302}$ | $0.371_{0.341-0.400}$ | $0.398_{0.368-0.428}$ |
| MedReason-8B | $0.407_{0.383-0.431}$ | $0.248_{0.224-0.275}$ | $0.331_{0.303-0.363}$ | $0.382_{0.353-0.412}$ |
| LLaMA-3.1-8B-Instruct | $0.451_{0.428-0.475}$ | $0.161_{0.138-0.184}$ | $0.281_{0.252-0.311}$ | $0.332_{0.304-0.360}$ |
| m1-7b-23k | $0.495_{0.440-0.551}$ | $0.155_{0.133-0.177}$ | $0.238_{0.211-0.264}$ | $0.291_{0.262-0.321}$ |
| Qwen-2.5-7B | $0.324_{0.301-0.347}$ | $0.174_{0.152-0.197}$ | $0.252_{0.223-0.279}$ | $0.287_{0.259-0.316}$ |
| *Fine-Tuned Models* | *Reasoning Recall* | *1-shot Acc.* | *5-shot Acc.* | *10-shot Acc.* |
| MedReason-8B (SFT) | $\mathbf{0.522}_{0.498-0.545}$ | $\mathbf{0.303}_{0.276-0.331}$ | $\mathbf{0.430}_{0.400-0.459}$ | $\mathbf{0.501}_{0.470-0.532}$ |
| LLaMA-3.1-8B-Instruct (SFT) | $0.485_{0.460-0.510}$ | $0.278_{0.249-0.307}$ | $0.411_{0.378-0.443}$ | $0.479_{0.448-0.509}$ |
| Qwen-2.5-7B (SFT) | $0.486_{0.461-0.510}$ | $0.249_{0.221-0.278}$ | $0.363_{0.335-0.394}$ | $0.425_{0.394-0.455}$ |

Table 3: Performance of models on `MedCaseReasoning`'s test set (N=897). Each model is evaluated on reasoning coverage and diagnostic accuracy (broken down into 1, 5, and 10-shot). Reasoning coverage is not available for *OpenAI o3* as the reasoning traces are not provided via API. We additionally perform supervised fine-tuning (SFT) on three open-source models on the training split of `MedCaseReasoning` and find significant improvements in both reasoning coverage and diagnostic accuracy.

We perform supervised fine-tuning (SFT) on two popular open-sourced models: *Qwen-2.5-7B-Instruct* and *LLaMA-3.1-8B-Instruct*. Additionally, to test the marginal value of our reasoning dataset, we include *MedReason-8B*, a model based on *LLaMA-3.1-8B-Instruct* that was previously fine-tuned on a synthetic medical reasoning dataset. We updated each model with full-weight fine-tuning for three epochs on 8 NVIDIA H100 GPUs with a learning rate of 2e-5 and batch size of 256. All other hyperparameters are default according to the *verl* Python package implementation of SFT (*version v0.3.0.rc0*).

## 3 RESULTS

### 3.1 VALIDITY OF GENERATED CASE PROMPTS, DIAGNOSTIC REASONINGS, AND FINAL DIAGNOSES

We performed a human validation of the model-generated case prompts, diagnostic reasons, and final diagnoses on the `MedCaseReasoning` dataset. Our study included four U.S. board-certified physicians reviewing a total of 100 cases. Each physician reported spending an average of 10-20 minutes per case. We found a high degree of agreement that the generated cases were free of hallucinations in the case prompt or diagnostic reasoning (98%). We also report that 92% of final diagnoses were reported to have been faithful to the article and reasonably inferrable from the details of the case prompt. Finally, 93% of diagnostic reasoning steps were faithful to the case report and clinically relevant. The results are found in the Appendix (Table 5).

### 3.2 DIAGNOSTIC ACCURACY

We evaluate the following combination of frontier reasoning models and popular open-sourced models: *OpenAI o3*, *DeepSeek R1*, *QwQ-32B*, *MedReason-8B*, *m1-7b-23k*, *LLaMA-3.1-8B-Instruct*, and *Qwen-2.5-7B-Instruct*.

The models are each evaluated on 1-shot, 5-shot, and 10-shot accuracy, listed in Table 3 and Table 6. Overall, we found that diagnostic reasoning remains a difficult task for top-performing LLMs. *OpenAI o3* performs significantly better than the rest of the models, with 64.5% 10-shot accuracy on the `MedCaseReasoning` test set, whereas *DeepSeek R1* achieves 48.0%. In comparison, *OpenAI o3* and *DeepSeek R1* achieve similar scores of 62.3% and 43.7% 10-shot accuracy on NEJM CPC, signaling the validity of `MedCaseReasoning` as an open-access alternative evaluation set for diagnostic cases.

We find that supervised fine-tuning significantly improves model performance across both test sets. For example, on the `MedCaseReasoning` test set, *MedReason-8B* improves by 31% 10-shot accuracy, outperforming *DeepSeek R1*. Of note, this model also improves by 18% on NEJM CPC and outperforms *QwQ-32B*. This provides evidence of the generalizability of `MedCaseReasoning` training data, as NEJM CPC consists of out-of-distribution, hand-crafted diagnostic cases that are often much longer in length.

### 3.3 Validity of Reasoning Recall Metric

We validate our LLM-as-a-judge for determining reasoning recall with verification from a board-certified physician. The physician was given N=33 cases and was asked to verify the LLM judge's decisions on a total of 89 pairs of groundtruth reasons and model thinking traces. The cases were randomly sampled across all evaluated models. For example, for a given case report that contained three reasoning steps, the physician was asked to cross-check each step against the entire model reasoning trace to see if it was considered. The physician found 84/89 (94.4, 95% CI of 84.8%-100%) of pairs of reasons and thinking traces to be correctly assessed by the model, and a total of 31/33 (93.9, 95% CI of 84.8%-100%) cases to be completely accurately assessed by the model.

### 3.4 Evaluation of Reasoning LLMs

We evaluate six of seven models that provide reasoning traces on reasoning recall and find that the top model, *DeepSeek R1*, covers 64.2% of the reasoning steps provided in case reports. We also found that *MedReason-8B*, a model explicitly trained on synthetically generated medical reasoning traces, did not have substantially higher reasoning recall, and even had lower recall than *LLaMA-3.1-8B-Instruct*'s base model. However, after SFT on `MedCaseReasoning`, each of the base models improved significantly on reasoning recall. For example, *MedReason-8B* improved by 28% and *Qwen-2.5-7B-Instruct* improved by 50%.

Common types of case report reasoning not found within the base model reasoning traces were either missing candidate diagnoses or exclusionary symptoms for common diagnoses. For example, one case report focused on a case of Adult-onset Still's disease (AOSD), which presented very similarly to Rocky Mountain Spotted Fever (RMSF). The model (*m1-7b-23k*) made a diagnosis for RMSF, missing reasoning about the patient presentation from the case report: "We feel that our patient's symptoms were not due to RMSF, because the rash was salmon colored and worsened with the fever spikes.". We found a significant correlation between model performance and reasoning recall (Pearson r=0.710, p=0.0485), indicating the value of measuring reasoning steps as a proxy for model performance itself. Furthermore, we also observe a significant correlation between the length of the model's reasoning trace and the reasoning recall (r=0.790, p=0.0196).

## 4 Discussion

Our work introduces `MedCaseReasoning`, the first open-access dataset designed to evaluate the alignment of LLMs with clinician-authored diagnostic reasoning. `MedCaseReasoning` addresses a critical gap in current medical benchmarks: the assessment of diagnostic accuracy without scrutinizing the underlying reasoning process. We report two key findings: first, even top-performing LLMs exhibit deficiencies in aligning with clinician reasoning, achieving a maximum recall of 64.2%. Second, fine-tuning models on the reasoning traces derived from `MedCaseReasoning` significantly improves both their recall with clinician reasoning and their

final diagnostic accuracy. These findings underscore the value of `MedCaseReasoning` in both evaluating and enhancing the clinical diagnostic capabilities of LLMs.

SFT with the `MedCaseReasoning` training dataset demonstrates that smaller models, such as *Llama 3.1 8B* and *Qwen 2.5 7B*, can achieve diagnostic accuracy comparable to or exceeding that of larger models like *Qwen1.5-32B* and *DeepSeek-R1* after SFT. While prior studies (Chen et al. (2024); Wu et al. (2025)) have explored learning reasoning from synthetic traces generated by more powerful models, our research is the first to demonstrate the efficacy of training directly on clinician-written diagnostic reasoning.

Compared to established benchmarks like MedQA, where leading models such as *GPT-4o* have already achieved over 90% accuracy, diagnostic performance on `MedCaseReasoning` currently peaks at 64.5%. This suggests `MedCaseReasoning` presents a more challenging task, focusing on the nuanced alignment with expert reasoning. This characteristic is shared with complex diagnostic case report datasets like the NEJM CPC; indeed, we observe a strong correlation in diagnostic performance between `MedCaseReasoning` and NEJM CPC (Figure 2). However, `MedCaseReasoning` offers distinct advantages: it is open-access, unlike the license-restricted NEJM CPC, and provides a substantially larger corpus, with nearly 14,489 examples (including a 13,092-example training set), compared to NEJM CPC's 302 test cases. Additionally, the `MedCaseReasoning` case curation pipeline is also extensible to other case reports, so the dataset is able to be extended as more reports become available, allowing the dataset to be updated to reflect current medical guidelines.

Our study presents several limitations. First, some case reports may lack sufficient detail for a definitive diagnosis or present trivial cases. The QA conversion process can also introduce variability, where case details may be inadvertently left out or hallucinated. While we implemented a clinician-validated filtering pipeline for the test set to ensure reasoning statements are grounded in the case presentation, some intractable cases (eg. ones where the diagnosis cannot be made without certain information) or trivial cases (eg. ones where the diagnosis is given away in the prompt) still persist. Second, `MedCaseReasoning` captures the case presentation at a single timestep before asking for a final diagnosis. It does not reflect the iterative, multi-stage nature of real-world clinical diagnosis, which involves refining differential diagnoses based on evolving information from tests, imaging, and treatment responses. Third, our reasoning recall metric only captures clinical reasoning provided within case reports. Diagnostic reasoning is inherently subjective, and while our extensive training corpus aims to encompass diverse diagnostic standards, the alignment metric should be interpreted as adherence to observed clinical reasoning patterns across a diverse range of clinicians rather than an absolute, single gold standard. Diagnosing rare and complex diseases has broad societal implications on patient health, patient-doctor interactions, and the trustworthiness of LLMs. Our study aims to shed light on a key factor mediating all three of these factors, namely diagnostic reasoning.

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

## 5 APPENDIX

| Question | Physician Agreement |
|---|---|
| Q1: There are no hallucinated details in the case prompt or diagnostic reasoning. | $98.0\%_{95.0-100.0\%}$ |
| Q2: The final diagnosis is faithful to the article and reasonably inferrable from the details in the case prompt. | $92.0\%_{86.0-97.0\%}$ |
| Q3: The diagnostic reasoning steps are faithful to the case report and are clinically relevant. | $93.0\%_{88.0-98.0\%}$ |

Table 4: Physician agreement rates with 95% confidence intervals for diagnostic prompt evaluation, with $N = 100$ total case reports reviewed. We find that diagnostic cases generated from case reports contain low rates of hallucinations and are faithful to the original reports.

| Journal Name | Prevalence (%) |
|---|---|
| Journal of Medical Case Reports | 8.20% |
| Clinical Case Reports | 7.57% |
| International Journal of Surgery Case Reports | 6.81% |
| Radiology Case Reports | 6.36% |
| Journal of Surgical Case Reports | 2.88% |
| JAAD Case Reports | 2.47% |
| SAGE Open Medical Case Reports | 2.16% |
| Journal of Orthopaedic Case Reports | 2.10% |
| Case Reports in Medicine | 1.87% |
| European Heart Journal: Case Reports | 1.60% |

Table 5: Top 10 medical case report journals by prevalence in the PMC Open Access Subset dataset. In total, there are 813 unique journals in `MedCaseReasoning`, spanning 30+ different medical specialties across multiple countries.

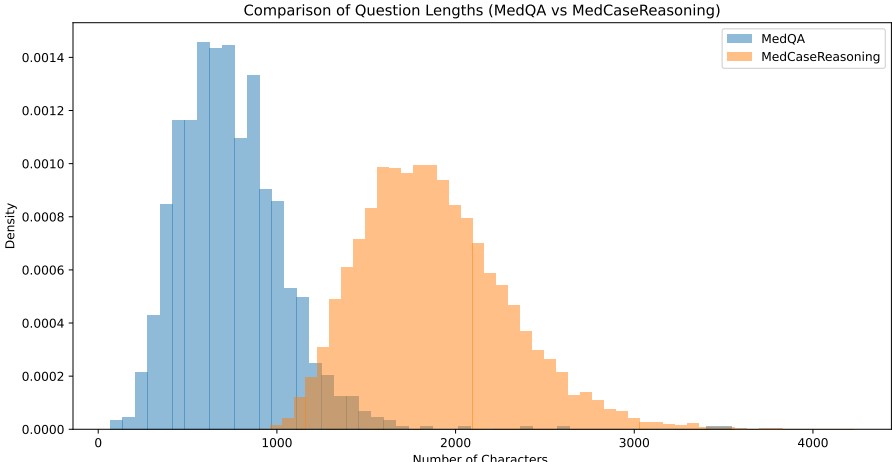

Figure 3: Comparison of length of questions from MedQA vs. `MedCaseReasoning`. Diagnostic case prompts are, on average, 2.5x longer in `MedCaseReasoning` and contain real patient information vs. synthetic case vignettes.

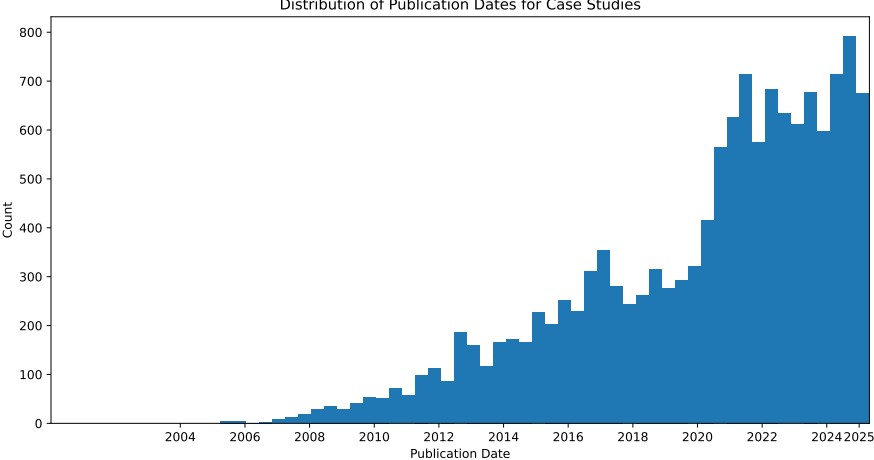

Figure 4: Distribution of dates of publication for PMC case reports used in `MedCaseReasoning`. Cases are largely from recent dates (after 2020), and over 500 cases are after Jan 1 2025.

| NEJM Clinicopathologic Conferences (CPC) - Generalization | | | |
|---|---|---|---|
| *Base Models* | *1-shot Acc.* | *5-shot Acc.* | *10-shot Acc.* |
| OpenAI o3 | $\mathbf{0.430}_{0.377-0.490}$ | $\mathbf{0.579}_{0.526-0.639}$ | $\mathbf{0.623}_{0.569-0.675}$ |
| DeepSeek R1 | $0.272_{0.225-0.318}$ | $0.407_{0.354-0.467}$ | $0.437_{0.384-0.497}$ |
| QwQ-32B | $0.182_{0.136-0.225}$ | $0.281_{0.232-0.334}$ | $0.341_{0.288-0.397}$ |
| MedReason-8B | $0.169_{0.126-0.215}$ | $0.262_{0.212-0.308}$ | $0.295_{0.245-0.348}$ |
| LLaMA-3.1-8B-Instruct | $0.070_{0.043-0.099}$ | $0.142_{0.103-0.182}$ | $0.212_{0.165-0.255}$ |
| m1-7b-23k | $0.099_{0.066-0.136}$ | $0.172_{0.136-0.215}$ | $0.209_{0.162-0.252}$ |
| Qwen-2.5-7B | $0.070_{0.043-0.099}$ | $0.109_{0.076-0.142}$ | $0.123_{0.086-0.159}$ |
| *Fine-Tuned Models* | *1-shot Acc.* | *5-shot Acc.* | *10-shot Acc.* |
| MedReason-8B (SFT) | $\mathbf{0.179}_{0.136-0.225}$ | $\mathbf{0.278}_{0.228-0.328}$ | $\mathbf{0.348}_{0.295-0.404}$ |
| LLaMA-3.1-8B-Instruct (SFT) | $0.169_{0.126-0.212}$ | $0.255_{0.205-0.305}$ | $0.308_{0.255-0.358}$ |
| Qwen-2.5-7B (SFT) | $0.129_{0.093-0.166}$ | $0.202_{0.159-0.248}$ | $0.252_{0.205-0.305}$ |

Table 6: To validate the generalization of our dataset, we evaluate models on NEJM Clinicopathologic Conference (CPC) cases (N=302), previously used in McDuff et al. (2025). We find that models fine-tuned on `MedCaseReasoning` also significantly improve on NEJM cases, which are considered a gold-standard in diagnostic case reports.

| Diagnostic Reasoning | Stitched Reasoning |
|---|---|
| 1. Benign soft-tissue tumors (lipoma or fibroma) were considered but deemed unlikely because "Lipomas and fibromas typically present as solid masses and are rarely associated with infection." | Lipomas and fibromas typically present as solid masses and are rarely associated with infection, making benign soft-tissue tumors such as lipomas or fibromas unlikely. |
| 2. A canal of Nuck cyst was considered but excluded because "The canal of Nuck represents a cystic lesion extending into the inguinal region, which was ruled out based on imaging." | The canal of Nuck represents a cystic lesion extending into the inguinal region, which was ruled out based on imaging. |
| 3. Bartholin's gland abscess was supported by pelvic imaging showing a contrast-enhancing fluid collection at the gland sites; therefore, a diagnosis of bilateral Bartholin's gland abscesses was made. | Pelvic imaging showed a contrast-enhancing fluid collection at the gland sites, supporting the diagnosis; therefore, a diagnosis of bilateral Bartholin's gland abscesses was made. |

Table 7: The model being fine-tuned (*LLaMA-3.1-8B-Instruct*) is used to convert the enumerated diagnostic reasoning into a cohesive reasoning trace, as shown above. The model is asked to stitch the reasoning without introducing any additional information.

| Type | Content |
|---|---|
| **Article Title** | Sebaceous carcinoma of the breast predominantly characterized by intraductal growth: a case report |
| **Journal** | Surgical Case Reports |
| **Article Link** | https://www.ncbi.nlm.nih.gov/pmc/articles/PMC7040145/ |
| **Case Prompt** | A 47-year-old Japanese woman with no breast symptoms was referred after screening mammography revealed clustered pleomorphic calcifications in the left breast. Her mother had bilateral breast and ovarian cancer. On examination, there was no palpable mass or nipple discharge. Serum tumor markers (CEA, CA15-3, NCC-ST-439, BCA225) were within normal limits. Breast ultrasound showed a $13\times12\times7$ mm irregular, hypoechoic mass with clear margins and internal high-echogenic foci in the left breast. MRI demonstrated a $14\times11\times12$ mm lesion with early arterial enhancement in the same region. PET-CT revealed focal uptake (SUVmax 3.54) in the left breast without abnormal uptake elsewhere. Core-needle biopsy of the mass showed ductal carcinoma in situ, nuclear grade 2, negative for ER, PgR, and HER2, with a Ki-67 labeling index of 32.4%. Given her family history, hereditary breast and ovarian cancer syndrome was considered, but the patient declined genetic testing. The patient elected to undergo total left mastectomy with sentinel lymph node biopsy. |
| **Diagnostic Reasoning** | 1. Consideration of invasive ductal carcinoma — "SC is characterized by lobular forms or nests of tumor cells that exhibit sebaceous differentiation, which distinguishes SC from invasive ductal carcinoma (IDC)."
2. Consideration of glycogen-rich clear cell carcinoma — "Glycogen-rich clear cell carcinomas typically become periodic acid-Schiff (PAS)-positive and therefore can be distinguished from SC cells because they do not produce glycogen." |
| **Final Diagnosis** | Sebaceous carcinoma |

Table 8: A single example from `MedCaseReasoning` of a case prompt, extract diagnostic reasoning, and final diagnosis.

## Prompt 1: Converting Case Reports to Diagnostic QA with Reasoning

**You are an expert clinician–educator.**
Your job is to **transform a published case report into a teaching diagnostic case** that medical students can work through step-by-step.
In terms of style, think of NEJM's Clinicopathologic Conferences ("Case Records of the Mass General Hospital") as a template.

---

**RULES (Read Carefully—No Exceptions)**

---

1. **Source Fidelity** – Extract facts *only* from the supplied case report.
   - Do **NOT** invent, embellish, or "smooth out" missing data.
   - Paraphrase narrative prose into concise bullets where helpful, but never add new facts.

2. **Structure the Teaching Case in Three Phases**
   *Case Presentation → Diagnostic Reasoning → Final Diagnosis*

3. **Use the XML Tags Exactly as Shown**
   - `<think>` …`</think>` – your hidden analytic notes (not visible to students).
   - `<case_prompt>` …`</case_prompt>` – the information given to students *before* they generate a differential.
   - `<diagnostic_reasoning>` …`</diagnostic_reasoning>` – numbered bullet reasons, each built as a full sentence followed by a **direct quote**.
   - `<final_diagnosis>` …`</final_diagnosis>` – **single disease/entity name only**, nothing more.

4. **What Goes Inside `<think>`**
   1. *Key points* – What makes this case non-trivial or pedagogically interesting? This should guide where the breakpoint should be.
   2. *Ideal breakpoint* – What details of the case presentation should you include and exclude so that students have enough data to reason, but no spoilers?
   3. *Author's analytic distinctions* – How did they separate the final diagnosis from look-alikes and other conditions?

5. **What Goes Inside `<case_prompt>`**
   - Present *only* the facts known **before** a working differential was made: chief complaint, HPI, vitals, physical exam, and early investigations.
   - Do not include references to Figure 1, Table 1, etc. directly. Summarize any imaging findings from what is given in the text.
   - Present the case in the order presented in the case report (e.g., physical labs before imaging, etc.)
   - Omit any wording that directly states or hints at the final diagnosis.
   - Present this as closely as possible to the style in which the case report is written.

6. **What Goes Inside `<diagnostic_reasoning>`**
   - Numbered list (1., 2., 3., …).
   - Include every alternative diagnosis mentioned in the article, along with reasons why it was considered and excluded, if provided.
   - Each entry: *concise summary of reason* ["direct quote from article"].
   - You can use ellipses (…) to shorten the quote if there are irrelevant details.
   - Quotes must justify why one possibility rose or fell in likelihood.

7. **What Goes Inside `<final_diagnosis>`**
   - **Single disease/entity name** (e.g., sarcoidosis).
   - No adjectives, punctuation, or explanatory text.

---

**OUTPUT TEMPLATE (copy exactly)**

---

[fontsize=] $<$think$>$ 1. [Core tension] 2. [Best breakpoint of case report, what to include and what to exclude] 3. [Key analytic distinctions between competing diagnoses (taken from case report)] $</$think$>$
$<$case$_p$rompt $>$ [$Your case presentation text, faithful to the report and stopping at the breakpoint$] < /case_prompt >$
$<$diagnostic$_r$easoning $> 1. Summarized reasoning—"Direct quote from article…" 2. Summarized reasoning—"Direct quote from article…" 3. … < /diagnostic_reasoning >$
$<$final$_d$iagnosis $> Disease Name < /final_diagnosis >$

---

**SUPPLIED CASE REPORT**

---

[fontsize=] $<$case$_r$eport $> case_report < /case_report >$

## Prompt 2: Grading the Quality of Case Reports

You are an expert medical educator tasked with evaluating case reports for their diagnostic-reasoning value. The goal is to find case reports that have a similar style to diagnostic teaching cases from medical textbooks.

### CASE REPORT EVALUATION RUBRIC

**» HOW TO USE**

1. Read the entire case once without scoring.
2. Re-read, taking notes.
3. Inside `<think>…</think>`, write the reasoning that leads you to each score.
4. Output only the five XML tags shown after the rubric—nothing else.

[fontsize=] +————————————————————————————————————-+ | 1. THOROUGHNESS OF CASE PRESENTATION (1–5 points) | | Look for: HPI, past history, meds, allergies, vitals, | | focused exam, labs, imaging, hospital course, outcome. | | 1 – Seriously deficient (identifiers only; no vitals) | | 2 – Major gaps (HPI + vitals OR exam, not both). | | 3 – Adequate (present but sketchy details). | | 4 – Very good (complete data, clear timeline). | | 5 – Exemplary (serial data course, high quality). | +————————————————————————————————————-+ | 2. EXPLICIT DIFFERENTIAL DIAGNOSIS (Yes / No) | | >=2 plausible alternatives? If yes → "Yes"; else → "No". | +————————————————————————————————————-+ | 3. DEPENDENCE ON INTEGRATIVE CLINICAL REASONING (1–5) | | Measures need to combine >=2 data points (hx, labs, etc) | | 1 – Trivial: lone clue gives answer. | | 2 – Minimal: one dominant clue. | | 3 – Moderate: must merge TWO findings. | | 4 – High: THREE+ clues; requires synthesis. | | 5 – Outstanding: stepwise, complex reasoning. | +————————————————————————————————————-+ | 4. TRANSPARENCY OF DIAGNOSTIC REASONING PROCESS (1–5) | | 1 – None (no rationale). | | 2 – Superficial (lists w/o "why"). | | 3 – Adequate (brief pivots). | | 4 – Detailed (stepwise, probabilities). | | 5 – Model (structured, addresses pitfalls). | +————————————————————————————————————-+ | 5. STATED FINAL DIAGNOSIS (Yes / No) | | Is diagnosis clearly named? Yes → "Yes"; else → "No". | +————————————————————————————————————-+

**Additional Rules:**

1. If the given article is not actually a case report, output "NA" for all scores.
2. Think through whether the final diagnosis can be reasonably deduced from the case presentation when determining the educational value in #3.

### OUTPUT TEMPLATE (leave tags exactly as written)

[fontsize=] $<think>$ ...your internal reasoning for each item... $</think>$ $<case_presentation_score>$ $[1-5]$ $</case_presentation_score>$ $<differential_diagnosis_score>$ $[Yes/No]$ $</differential_diagnosis_score>$ $<integrative_reasoning_score>$ $[1-5]$ $</integrative_reasoning_score>$ $<transparency_score>$ $[1-5]$ $</transparency_score>$ $<final_diagnosis_score>$ $[Yes/No]$ $</final_diagnosis_score>$

### SUPPLIED CASE REPORT

[fontsize=] $<case_report>$ $case_report$ $</case_report>$

---

## Prompt 3: Detecting Hallucinations and Inconsistencies in Generated Case Prompts

**You are the Editor.**
Your job is to audit a draft teaching case that was produced from a published case report.
You must confirm strict compliance with all instructions, detect hallucinations, and ensure pedagogic quality.

---

### YOUR INPUTS

---

1. The original draft you must audit appears between
   `<generated_case>` …`</generated_case>`.

2. The source article appears between
   `<case_report>` …`</case_report>`.

Here is the original guidelines the draft was produced in accordance with:
`{convert_case_report_prompt}`

---

### CHECKLIST — FAIL ANY ITEM → RAISE A FLAG

---

**A. Source Fidelity**   ☐ Every fact in each section is traceable to the source article.
   ☐ No invented details or embellishments.

**B. Case Presentation Quality**   ☐ All facts from the case prompt are present in the source article.
   ☐ Contains only information known *before* the clinicians formed a differential.
   ☐ Does **not** reveal the final diagnosis (there should be room for at least some inference).
   ☐ Provides *sufficient* data (HPI, vitals, exam ± initial tests) for clinicians to formulate a reasonable differential and get the correct final diagnosis.

**C. Diagnostic Reasoning Section**   ☐ Each numbered entry starts with a summary of the reasoning **plus** a direct quote from the article.
   ☐ Quotes are verbatim or use ellipses (…) without changing meaning.
   ☐ Paraphrased quotes are okay, as long as they retain the original meaning.
   ☐ **Rationales reference only information that already appears in `<case_prompt>`** (not based on new findings, confirmatory tests, or data withheld from students).

**D. Final Diagnosis Tag**   ☐ Final diagnosis is *reasonably* deducible from the case-presentation facts.
   — i.e., the final diagnosis should not depend entirely on some test, imaging, or lab result not given in the case presentation.

**E. No Hallucinations Anywhere**   ☐ Every datum, quote, or diagnosis is found in the case report.

---

### HOW TO REPORT YOUR FINDINGS

---

Output **only** the two XML blocks below.

1. `<flags>` …`</flags>`
   • If an item fails, add a line `FLAG: [short descriptor]`.
   • Use one line per failed item, drawn from this controlled vocabulary:
   `CASE_PROMPT_HALLUCINATION, FINAL_DIAGNOSIS_IN_CASE_PROMPT,`
   `INSUFFICIENT_INFO_FOR_DIAGNOSIS, DIAGNOSTIC_REASONING_HALLUCINATION, OTHER.`
   • If **no** issues, write `NONE`.

2. `<editor_comments>` …`</editor_comments>`
   • Briefly justify each flag (one sentence each).
   • If no flags, you may omit or leave empty.

*Example when problems exist:* [fontsize=] $<$flags$>$ FLAG: $SOURCE_{F}IDELITY FLAG$ : $REASONING_{E}XTRA_{I}NFO$ $<$ $/flags$ $><$ $editor_{c}omments$ $>$ $-SOURCE_{F}IDELITY$ : $Mentions "family history of SLE," not present in article. - REASONING_{E}XTRA_{I}NFO$ : $Rationale cites a biopsy result that is not included in the case_{p}rompt. < /editor_{c}omments >$
*Example when everything passes:* [fontsize=] $<$flags$>$ NONE $</$flags$>$ $<$editor_{c}omments$ ><$ $/editor_{c}omments$ $>$

---

### INPUT BLOCKS TO REVIEW

---

Here is the reference case report: [fontsize=] $<$case_{r}eport > case_{r}eport < /case_{r}eport >$

Here is the diagnostic case generated by the model: [fontsize=] $<$case_{p}rompt > generated_{c}ase_{p}rompt < /case_{p}rompt >< diagnostic_{r}easoning > generated_{d}iagnostic_{r}easoning < /diagnostic_{r}easoning >< final_{d}iagnosis > generated_{f}inal_{d}iagnosis < /final_{d}iagnosis >$

## Prompt 4: Stitching Reasoning Trace From Enumerated Diagnostic Reasoning

```
Your job is to convert a list of diagnostic reasoning points into a cohesive diagnostic reasoning narrative.

INPUTS
* CASE_PROMPT:
{case_prompt}

* REASONING_POINTS (in the order they were generated):
{reasoning_points}

* FINAL_DIAGNOSIS:
{final_diagnosis}

TASK
Write a single, cohesive reasoning trace by stitching together all the REASONING_POINTS.
The REASONING_POINTS are given as an enumerated list, where each point is a brief summary of a particular
    reasoning point, followed by a quote from the full case report that supports the reasoning point.
This should be written from the perspective of a reasoning trace that an LLM chatbot would write in response to a
    case presentation.
Do not use the past-tense nature of REASONING_POINTS, instead make them present-tense and third-person as you are
    considering each potential diagnosis.

OUTPUT
Only use the reasoning from the REASONING_POINTS to come up with the final diagnosis.
Use ALL of the REASONING_POINTS in producing the reasoning trace.
Do not include your own reasoning over the case, only use the reasoning points provided.
Try to incorporate as much of the quotes as possible into the reasoning trace, using word-for-word copies when
    possible (don't actually put quotation marks in the reasoning trace).
You may rephrase the reasoning points, but only for style and tone, not for substance.
No headings, no bullets, no numbered -listsjust a continuous explanatory narrative.

Place this between the tags <stitched_reasoning> and </stitched_reasoning>
```

## Prompt 5: Grading Reasoning Recall

```
You are an experienced medical expert tasked with comparing diagnostic reasoning statements that support a given
    diagnosis for a given patient case.
Your goal is to find supporting statements in the predicted diagnostic reasons that match the groundtruth
    diagnostic reasons.

For each of the statements in Groundtruth Diagnostic Reasons, you need to find the statement or statements in the
    Predicted Diagnostic Reasons that state the equivalent justification for the diagnosis.
For instance, if the groundtruth diagnostic reason is "The patient has a fever", and the predicted diagnostic
    reason is "The patient has a fever due to a viral infection", then this is a match.
If the groundtruth diagnostic reason is "The patient has a fever", and the predicted diagnostic reason is "The
    patient has a sore throat", then this is not a match.

Instructions:
1. Analyze each statement in Groundtruth Diagnostic Reasons.
2. For each statement in Groundtruth Diagnostic Reasons, find any matching statements in Predicted Diagnostic
    Reasons.
3. Create a JSON object with the following structure:
   - The main key should be "matching_dict"
   - Each key within "matching_dict" should be a number representing a statement from Groundtruth Diagnostic
       Reasons
   - The value for each key should be a list of matching statements from Predicted Diagnostic Reasons
   - If there are no matches for a statement, use an empty array

Before providing your final output, wrap your analysis inside <diagnostic_comparison> tags:
1. List all statements from Groundtruth Diagnostic Reasons and Predicted Diagnostic Reasons.
2. For each statement in Groundtruth Diagnostic Reasons, consider potential matches from Predicted Diagnostic
       Reasons:
   - List pros and cons for each potential match
   - It's OK for this section to be quite long
3. Summarize your final matching decisions
4. In the JSON output, only include the statements that are in the Predicted Diagnostic Reasons.
5. In the JSON output, the statements should appear exactly as they are in the Predicted Diagnostic Reasons,
       verbatim, letter for letter. Do not modify the statements in any way, such as rewording them, adding
       punctuation, quotes, etc.

Wrap your JSON output in ```json tags.

Example of the required JSON structure:
```json
{{
  "matching_dict": {{
    "1": [],
    "2": ["Matching statement 1", "Matching statement 2"],
    "3": ["Matching statement 3"]
  }}
}}
```

### Prompt 6: Diagnostic Question Template

```
Read the following case presentation and give the most likely diagnosis.
First, provide your internal reasoning for the diagnosis within the tags <think> ... </think>.
Then, output the final diagnosis (just the name of the disease/entity) within the tags <answer> ... </answer>.

----------------------------------------
CASE PRESENTATION
----------------------------------------
{case_presentation}

----------------------------------------
OUTPUT TEMPLATE
----------------------------------------
<think>
...your internal reasoning for the diagnosis...
</think>
<answer>
...the name of the disease/entity...
</answer>
"""

a_prompt = """<think>
{reasoning}
</think>

<answer>
{answer}
</answer>
```

### Prompt 7: Diagnostic Accuracy LLM-as-a-judge

```
Is our predicted diagnosis correct (y/n)?
Predicted diagnosis: {predicted_diagnosis}, True diagnosis: {actual_diagnosis}
Answer [y/n].
```

