# OpenReview forum: "MedCaseReasoning: Evaluating and learning diagnostic reasoning from clinical case reports"
_ICLR.cc/2026/Conference — ICLR 2026 Conference Withdrawn Submission_

### Official Review · Reviewer_34z9 · 2025-10-28

**Soundness:** 2
**Presentation:** 2
**Contribution:** 2
**Rating:** 2
**Confidence:** 4

**Summary:**

This study addresses a significant gap in the evaluation of medical LLMs by focusing not only on diagnostic accuracy but also on the clinical reasoning process. The introduction of the MedCaseReasoning dataset, built from real-world cases, is a valuable contribution that aligns well with practical clinical scenarios.

**Strengths:**

This is a highly meaningful research direction, as it promotes a more comprehensive evaluation of LLM capabilities in the medical domain. The use of real clinical cases enhances the practical relevance and potential impact of the dataset.

**Weaknesses:**

However, as a dataset construction paper, the study has several notable shortcomings:

(1) Unclear and Confusing Description of Dataset Construction:

+ In Section 2.1 (Data Curation), the "Candidate Selection" process is poorly explained and lacks clarity. Additionally, the 1–5 scoring scale mentioned in this section is not clearly defined. The prompts included in the appendix are also confusing and difficult to interpret.

+ The overall description of the dataset is insufficient. Key details—such as the number of diagnostic types included, the distribution of reasoning steps, and other structural characteristics—are missing. These are critical for understanding the dataset's scope and for enabling its broader adoption.

(2) Poorly Explained Evaluation Methods:

+ In Section 2.1, the authors state: "As expert verification on each step is prohibitively expensive, we perform our clinician validation only on the outputs of the last step." It is unclear how this approach aligns with later sections that refer to full reasoning trace evaluation, creating confusion about which steps were actually validated.

+ In Section 2.3 (Evaluating Reasoning Traces in Models), the authors use an LLM-as-a-judge approach, where o4-mini is prompted to return a JSON indicating whether ground-truth reasons are present in the model's reasoning trace. The evaluation criteria are vague: Is it based on exact match, keyword presence, or semantic similarity? This needs clarification.

(3) Insufficient Experimental Analysis:

+ The evaluation of the dataset is limited to few-shot and supervised fine-tuning (SFT) settings. A more thorough analysis—such as performance across different reasoning step lengths, case complexities, or diagnostic categories—would better demonstrate the dataset's characteristics and utility.

+ In Section 2.4, the authors mention using an LLM to "stitch" extracted reasoning points into a coherent trace. The justification for this method is unclear, and no ablation study is provided to compare this approach with alternative formatting strategies.

(4) Other Issues:

+ There is an inconsistency in the data split numbers: Figure 1 mentions "897 test cases and 13,092 training cases," which does not match the total of 14,489 cases mentioned earlier.

+ In Section 3.3 (Validity of Reasoning Recall Metric), the origin of the "89 pairs of groundtruth reasons" is not explained.

**Questions:**

None

---

### Official Review · Reviewer_6Ki7 · 2025-10-30

**Soundness:** 2
**Presentation:** 2
**Contribution:** 2
**Rating:** 4
**Confidence:** 4

**Summary:**

The authors introduce MedCaseReasoning, an open-access benchmark and training corpus built from 14489 clinician-authored case reports (with 897 test cases) to evaluate both diagnostic accuracy and alignment with clinician reasoning. They define a "reasoning recall" metric that checks whether a model's explanation covers the enumerated reasons extracted from the source case. Using this benchmark, frontier and open models underperform on both diagnosis and reasoning alignment. Supervised fine-tuning on MedCaseReasoning's reasoning traces boosts both accuracy and recall, and gains transfer to NEJM CPC cases.

**Strengths:**

- This paper focuses on a new task about reasoning alignment grounded in clinician-authored case reports. It provides large, open-access corpus spanning 800+ journals and 30+ specialties.
- A "reasoning recall" metric is proposed to evaluate the reasoning quality of the model outputs compared to constructed references.
- Experiments include a broad range of LLMs, showing that baseline models struggle on this benchmark.
- SFT on reasoning traces brings consistent improvements in both diagnostic accuracy and reasoning recall, and these improvements transfer to NEJM CPC.

**Weaknesses:**

- The data construction heavily relies on LLMs, raising concerns about bias and hallucination in generated content. Although manual evaluations were conducted by physicians on a small subset, the performance of o4-mini, which generated the diagnostic QA pairs, is not reported on the final MedCaseReasoning test data.
- The reasoning metric measures only recall, not precision. As a result, a model may "over-explain" by adding spurious or incorrect reasons without affecting the evaluation scores.
- The reasoning recall is computed by selecting the best response (if any) among 10 generated samples, which may overestimate true recall. Reporting the average recall across samples and discussing whether recall correlates with diagnostic accuracy would strengthen the analysis.
- While MedQA and MedXpertQA are labeled as having "no expert reasoning," their questions still require reasoning by LLMs. It remains unclear whether LLMs fine-tuned on MedCaseReasoning can generalize to other medical reasoning tasks such as MedQA or MedXpertQA.

**Questions:**

1. The dataset construction heavily relies on LLM-generated outputs. How do the authors ensure that these generated diagnostic QA pairs are free from bias and hallucination? Was the performance of o4-mini evaluated on the final MedCaseReasoning test set?
2. The reasoning metric currently measures only recall. Have the authors considered incorporating precision (or F1) to penalize spurious or incorrect reasoning steps that might inflate recall?
3. Reasoning recall is reported using the best of 10 generated responses. Could this setup overestimate true model performance? Would the authors provide the average recall across all generations, and discuss whether reasoning recall correlates with diagnostic accuracy?
4. While MedQA and MedXpertQA are marked as datasets without "expert reasoning," their questions still require reasoning from LLMs. How well do models fine-tuned on MedCaseReasoning generalize to such medical reasoning benchmarks that lack explicit rationales?

---

### Official Review · Reviewer_9KKc · 2025-11-01

**Soundness:** 2
**Presentation:** 2
**Contribution:** 2
**Rating:** 4
**Confidence:** 3

**Summary:**

This paper introduces an open-access benchmark termed MedCaseReasoning and also a corpus to evaluate diagnostic accuracy and alignment with clinician reasoning. The paper defines a metric to check if a model's explanation covers the enumerated reasons extracted from the source. Based on this benchmark, the paper evaluated the models underperforming on both diagnosis and reasoning alignment.

**Strengths:**

This paper’s main contribution is the creation of a large, open-access benchmark that explicitly ties diagnostic answers to clinician-written reasoning, moving beyond accuracy-only evaluations that dominate MedQA/MMLU-style datasets. The authors assemble 14489 QA cases from PMC case reports via a multi-stage, clinician-validated pipeline and release a curated 897-case test split, which makes the task both realistic and unsaturated. The suite of results is also compelling: models perform far from ceiling (e.g., OpenAI o3 at 65% 10-shot on the test set) and the proposed training use of the dataset yields sizable, documented gains in both diagnosis and alignment with clinician rationales, with performance correlating across MedCaseReasoning and an external NEJM CPC set.

**Weaknesses:**

The data curation and evaluation loops lean heavily on LLMs (o4-mini for QA/trace extraction and scoring; Gemini-2.5-pro for quality checks; GPT-4-class models for judging), which raises concerns about hidden biases, propagation of subtle hallucinations, and circularity, especially given that only 100 cases received clinician spot-checks and that "pass/fail" filtering depends on model judgments.

The central metric (reasoning recall against clinician bullets) captures coverage but not precision (spurious or incorrect steps), internal consistency, or causal structure. As a result, models could game the score by producing longer traces, a risk the paper acknowledges only indirectly via correlation with trace length.

The reliance on LLM-as-a-judge to score diagnostic correctness also introduces measurement uncertainty, and comparisons are asymmetric because some frontier models do not expose full traces, preventing apples-to-apples reasoning evaluation.

Finally, the benchmark focuses on single-snapshot diagnostic vignettes. It is unclear how gains from SFT on these traces transfer to multi-stage clinical workflows or to adjacent tasks (e.g., MedQA/MedXpertQA) beyond the NEJM CPC correlation.

**Questions:**

Please address the weaknesses.

---

### Official Review · Reviewer_Vven · 2025-11-01

**Soundness:** 1
**Presentation:** 1
**Contribution:** 2
**Rating:** 2
**Confidence:** 3

**Summary:**

The paper presents a dataset for evaluating LLMs’ medical reasoning capability, together with reasoning traces. The dataset originates from the PMC Open Subset, and case reports are filtered so that all selected case reports come with differential diagnosis, come with final diagnosis, etc. The paper seems to claim that the evaluation of the reasoning process. The dataset can be used for fine-tuning LLMs.

**Strengths:**

1. The dataset may be potentially useful.

**Weaknesses:**

1. I don’t see what the diagnostic reasoning (or reasoning traces) is. This idea should be fully explained/defined, as it seems important for the paper.
2. I’m not sure about this, but it looks like the paper uses differential diagnosis as reasoning traces. From the terms “diagnostic reasoning” or “reasoning traces,” it sounds like the deduction process. Meanwhile, differential diagnosis looks like a comparison between possible diseases. It would be nice if the paper could provide if they can be seen as the same. Otherwise, the paper should explain why differential diagnosis can be used to evaluate the “diagnostic reasoning.”

Because the core part of the paper is not fully explained, I have to rate the paper negatively.

**Questions:**

I want to see some discussions on the weaknesses.

---

### Note · Authors · 2026-01-09

I have read and agree with the venue's withdrawal policy on behalf of myself and my co-authors.